# A Study on the Effect of $O_2$ Diffusion on the Retention Time of Inert Agents

**GoonHo Kim, Ju-Hong Cha, Jee-Hun Jeong and Ho-Jun Lee \***

Department of Electrical Engineering, Pusan National University, Pusan 46241, Korea;
windthink@pusan.ac.kr (G.K.); ciwsssh@nate.com (J.-H.C.); hemn4221@gmail.com (J.-H.J.)

**\*** Correspondence: hedo@pusan.ac.kr; Tel.: +82-51-510-2301



**Featured Application: The approach proposed in this paper can be applied in the field of fire safety in industrial facilities where water extinguishing agents are not available. In particular, it can be useful for the design and maintenance of gas fire extinguishing systems. Accurate analysis of the $O_2$ diffusion in inert agents provides an accurate prediction time (retention time) to prevent recurrence of fire.**

**Abstract:** Gaseous agents are widely used in fire extinguishing systems (FESs) when water extinguishing agents are unavailable. The extinguishing ability of the FES-gaseous agent is determined by the retention time (hold time) at which its concentration is maintained. In particular, the retention time of the inert agent is determined by the $O_2$ inflow from the outside. However, current theoretical models for inert agents do not provide an accurate model for the diffusion of incoming $O_2$. Specifically, because the theoretical equations do not include $O_2$ diffusion or include too large a value, there is a large difference between the measured and theoretical retention times. Therefore, in this study, accurate $O_2$ diffusion was verified through experimental and numerical analyses using three types of deactivators and reflected in the existing theoretical model. $O_2$ diffusion was analyzed through the interface slope $\alpha$ and diffusion velocity $v_d$. As a result, this proposed method can predict the retention time more accurately than existing theoretical models.

**Keywords:** fire extinguishing system; inert agent; retention time; hold time; diffusion flux

## 1. Introduction

The most important aspect to consider when using gaseous agents in a fire extinguishing system (FES) is the ability of the system to maintain the concentration of the agents in the enclosure for an extended period of time [1,2]. Therefore, when designing an FES-gaseous system, the structure of the space to be protected, the sealing ability of the agent in the space to be protected, and the design concentration of the agent used, among others, must be considered. The most useful way to evaluate the capabilities of this FES-gaseous system is to measure its retention time or hold time. The retention time is defined as the time until the concentration of the FES-gaseous agent drops below the specified threshold at the equivalent specified height for a protected room. At this time, the specified height is usually set as the point of maximum combustibles or 75% to 80% of the maximum height of the enclosure, and the specified threshold is set to approximately 80% of the designed concentration [1–4]. In particular, the extinguishing ability of the inert agent is the ability of the $O_2$ concentration in the protected space to maintain a nonflammable concentration that cannot be burned. Therefore, as the concentration of the inert agents refers to the concentration of $O_2$, the retention time of the inert agents can eventually be determined by the threshold of the concentration of $O_2$ [5].

The design standards for inert agent systems published by the National Fire Protection Association 2001 (NFPA 2001) [6] and the International Standards Organization 14520-1 (ISO 14520-1) [7] provide simplified physical models that can predict retention time. These models are well-established theories for orifice flow, which model the decrease in the agent concentration assuming the worst case in the space to be protected where the FES-gaseous system is designed [1–4,8–11]. However, retention time predictions using these simplified models are often inaccurate; namely, they present an overly optimistic or conservative approximation of the retention time [3]. Previous studies have shown that the NFPA and ISO models provide almost the same input data to analyze output data; however, the retention time prediction using the NFPA model differs by up to twice that of the ISO model [3,4,12]. This is because when interpreting the models, it is assumed that there is no species diffusion (NFPA2001, *the sharp descending interface model*), or that species diffusion is rapid (ISO 14520-1, *the wide descending interface model*). Thus, species diffusion is a significant factor for the retention time. To solve this problem, Hetrick [3,4] proposed a new model (*the thick descending interface model*) by measuring the interface thickness by species diffusion. The assumption for species diffusivity in this model is reformulated as a combination of *the sharp descending interface model* and *the wide descending interface model*. However, in this model, the theoretical equation for the evaluation of retention time reflects the halocarbon-compound agent and inert agent according to the same criteria.

The purpose of this study is to improve the accuracy of retention time predictions when using an inert agent. The $O_2$ diffusion, which affects the retention time in an inert agent, is measured, and a new method is proposed to reflect the measured $O_2$ diffusion in the theoretical equation for the retention time. The proposed method improves the accuracy of the retention time predictions by including the diffusion flux caused by the difference in concentration of $O_2$, in addition to the advection flux generated by the difference in density between the air–agent mixture gas inside the enclosure and the air outside. To reflect the diffusion flux, we define the interface slope $\alpha$ and interface thickness $\omega$. The interface slope $\alpha$ is defined to reflect the diffusion of the $O_2$ that is not included in the existing theoretical equation, and indicates how the interface thickness $\omega$ and retention time $t_r$ are related. The interface slope $\alpha$ and the diffusion velocity $v_d$ related to transport by species diffusion (diffusion flux) were measured through experiments. An additional numerical analysis is performed to obtain information that is difficult to obtain through experiments.

In this study, three inert agents, IG-01, IG-55, and IG-541 (except IG-100), were used to confirm the newly proposed method. The material properties of each inert agent are listed in Table 1.

**Table 1.** Type and material characteristics of inert agents.

| Agent | $N_2$ (%) | Ar (%) | $CO_2$ (%) | Molecular Mass (g/mol) |
|-------|-----------|--------|------------|------------------------|
| IG-01 | 0 | 100 | 0 | 39.948 |
| IG-55 | 50 | 50 | 0 | 33.980 |
| IG-541 | 52 | 40 | 8 | 34.066 |
| IG-100 | 100 | 0 | 0 | 28.013 |

As with the existing theory, transients caused by agent discharge such as cooling effects and enclosure implosions are omitted [3,4,13–17].

## 2. Theoretical Background

### 2.1. Theoretical Models for Retention Time

For an inert agent, the retention time can be determined by various parameters such as temperature, pressure, density, agent concentration, leakage area size and location, and agent discharge conditions. It is challenging to predict the retention time considering all these variables. Therefore, the theoretical models for the retention time of an inert agent minimize the following variables to simplify the prediction [1–4,13–17].

- Agent discharge steps are not considered.
- The protected room has a constant cross-sectional area according to its height.
- Leakages through the protected room boundaries occur only at the upper and lower extremes of the room elevation.
- The internal and external environments of the protected room have a standard temperature and pressure.
- All species diffusivity is assumed to be the same or ignored.
- The thermal effects are ignored.
- The initial state of the agent in the protected room is a homogeneous mixture.

Simplification through these assumptions may reduce the accuracy of the retention time, but because it improves the accessibility of the analysis, the designer or manager of the fire extinguishing equipment in the enclosure has the advantage of easily predicting the retention time.

The models for the retention time based on these assumptions can be classified into four models, as shown in Figure 1.

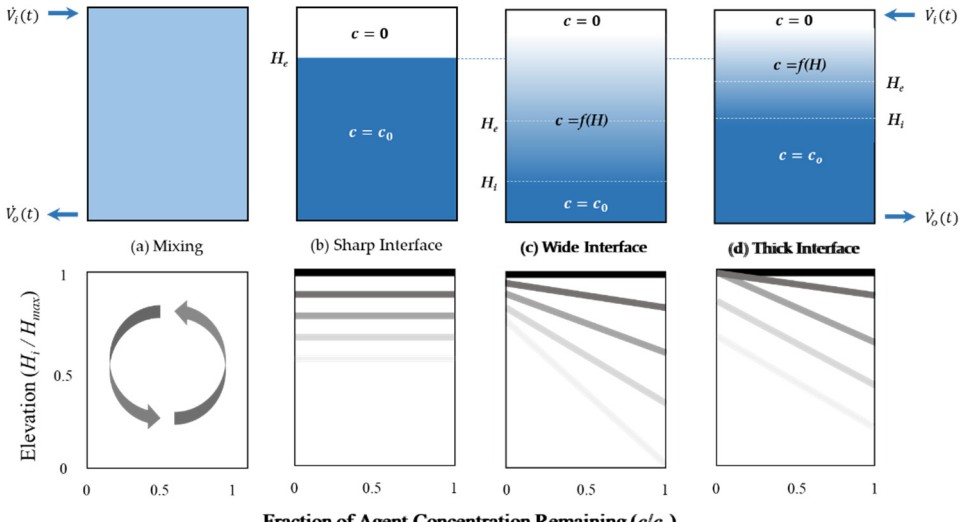

$c$ : Concentration of inert agents    $c_0$ : Initial concentration of inert agents    $H_{max}$ : Maximum height of the enclosure    $H_i$ : The leading edge of the interface height
$H_e$ : Equivalent specified height of the enclosure    $\dot{V}_i(t)$ : Inflow volumetric flow    $\dot{V}_o(t)$ : Outflow volumetric flow

**Figure 1.** Schematic diagrams of the theoretical models for the retention time: (**a**) The continuous mixing model; (**b**) the sharp descending interface model; (**c**) the wide descending interface model; (**d**) the thick descending interface model.

- *The continuous mixing model:* NFPA 2001, Annex C, Figure 1a
- *The sharp descending interface model:* NFPA 2001, Annex C, Figure 1b
- *The wide descending interface model:* ISO 14520-1, Annex E, Figure 1c
- *The thick descending interface model:* Considered by Hetrick [3,4], Figure 1d

Figure 1 shows the agent concentration for the enclosure height. Therefore, it is expressed in four models depending on the degree of diffusion of $O_2$ flowing from the outside.

When the density $\rho_{mix}$ of the air–agent mixture gas inside the enclosure is greater than the density $\rho_{air}$ of the air surrounding the enclosure, the air–agent mixture gas flows out through the leakage area at the bottom at a volumetric flow rate of $\dot{V}_o(t)$, and the outside air flows in through the leakage area at the top at a volumetric flow rate of $\dot{V}_i(t)$. (If $\rho_{mix}$ is smaller than $\rho_{air}$, such as IG-100, air flows in from the lower part and the air–agent mixture gas flows out from the upper part.) At this time, an interface is formed between the incoming air and the air–agent mixture gas inside the enclosure.

Therefore, the theoretical models are classified according to the degree of mixing between air and air–agent mixture gas, that is, the size of the interface [1–4].

*The continuous mixing model* can be applied when an inert material such as the IG-100 has a small difference in density inside and outside the enclosure; furthermore, the incoming $O_2$ diffuses throughout the enclosure. However, because of the insignificant difference in the internal/external density, very few $O_2$ are introduced inside the enclosure. Therefore, the retention time for this model tends to be very long. This model is not applicable to gaseous agents, except for IG-100. Therefore, because this is a model to which the existing theoretical equation is not applied, experiments and numerical analyses on IG-100 were not performed in this study.

*The sharp descending interface model* assumes no transport by species diffusion. As shown in Figure 1b, the interface slope $\alpha$ and interface thickness $\omega$ are zero because the air flowing from the top of the enclosure does not mix with the internal agent–air gas mixture. That is, it is assumed that the retention time is determined only by fluid flow transport. Therefore, this model results in an overly optimistic prediction of retention time.

*The wide descending interface model* includes transport by species diffusion and fluid flow. It is assumed that the incoming air is mixed with the internal agent–air gas mixture at a known proportion [3,7]; half of the incoming air is mixed with the agent–air gas mixture, and the other half remains at the top. Therefore, the concentration of the agent decreases linearly from the leading edge of the interface, $H_i$, as shown in Figure 1c. This model results in a shorter retention time than that obtained by *the sharp descending interface model* because it includes transport by species diffusion. However, it significantly reflects the species diffusion, namely, the interface thickness, which increases over time and finally appears over the entire height of the enclosure. The interface slope $\alpha$ and thickness $\omega$ are determined by the concentration of the inert agent and the height of the enclosure [3]. Therefore, the retention time was evaluated conservatively.

The two aforementioned models present extreme results of the theoretical retention time; therefore, *the thick descending interface model* compensates for these shortcomings [3,4]. It reflects the species diffusion (the interface thickness) measured by experiments; it is composed of the recombination of *the sharp descending interface model* and *the wide descending interface model*. As shown in Figure 1d, *the thick descending interface model* assumes that the maximum interface thickness is formed after a certain period of time, after which the interface is maintained at a constant thickness and moves under the enclosure over time. In this model, the degree of species diffusion is expressed as a dimensionless interface thickness, which is expressed as a function of the enclosure height. However, species diffusion is caused by differences in temperature, concentration, pressure, etc. at the boundary of the species facing each other, and it has been shown that the influence on the height of the enclosure will not be significant [12].

The retention times of the four models mentioned here are compared in Figure 1; the specified height $H_e$, which determines the retention time of the b, c, and d models, indicates the value at the same time $t = t_r$. As shown in Figure 1, *the sharp descending interface model* indicates the slowest retention time and *the wide descending interface model* indicates the fastest. *The thick descending interface model*, which complements the two models, indicates the retention time between the values of the other models.

### 2.2. Existing Governing Equation

The theoretical equation for the retention time is derived based on mass conservation through *the sharp descending interface model* and is extended to *the wide descending interface model* and *the thick descending interface model* [1–4,8–11]. This is because the analysis becomes significantly complicated when species diffusion is added to the retention time. Figure 2 shows a schematic diagram of the $O_2$ concentration distribution profile for *the sharp descending interface model*.

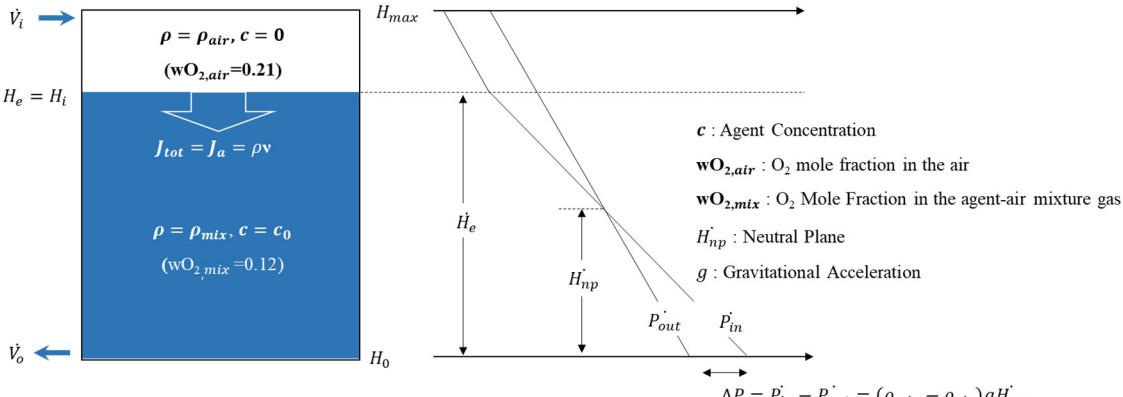

**Figure 2.** Schematic diagram of the $O_2$ concentration distribution of *the sharp descending interface model*.

From Figure 2, the theoretical equation for the retention time can be calculated. The theoretical equation for the retention time is a function of height, where a one-dimensional analysis is performed in the enclosure; proof of this analysis has already been presented in several studies [1–4,9,17]. In the case of Figure 2, the density $\rho_{mix}$ of the air–agent mixture gas inside the enclosure is greater than the density $\rho_{air}$ of the air surrounding the enclosure. If $\rho_{air}$ is larger than $\rho_{mix}$, the direction of the inflow and outflow volumetric flow is reversed to the above Figure 2.

Fresh air introduced into the enclosure by the difference between the density $\rho_{mix}$ of the air–agent mixture gas inside the enclosure and the density $\rho_{air}$ of the external air moves to the bottom of the enclosure along the z-direction by the total flux $J_{tot}$. At this time, the total flux $J_{tot}$ can be expressed by Equation (1) as follows [12,17,18]:

$$J_{tot} = J_a + J_d \tag{1}$$

When analyzing the retention time upon applying *the sharp descending interface model*, which does not reflect species diffusion, Equation (1) is interpreted as $J_{tot} = J_a$ by reflecting $J_d$ as zero. Therefore, when the retention time is calculated by changing the z-direction advection flux $J_a$ to a function of the specified height $H_e$, it is expressed as shown in Equations (2) and (3) [3]:

$$J_{tot} = J_a = v_a \varphi_{mix} = \varphi_{mix} \frac{\partial \dot{H}_e}{\partial t} = -\varphi_{mix} \frac{\dot{V}_o}{A_f} \tag{2}$$

$$\frac{\partial \dot{H}_e}{\partial t} = -\frac{\dot{V}_o}{A_f} \tag{3}$$

In the aforementioned equations, if $\dot{V}_o$ is converted to a function of $\dot{H}_e$ and both sides are integrated, the theoretical retention time can be derived as shown in Equation (5):

$$\dot{V}_o = A_o C_d C_u \left( \frac{2\Delta P_o}{\rho_{mix}} \right)^n = A_o C_d C_u \left\{ \left( \frac{2}{\rho_{mix}} \right) \left[ \frac{(\rho_{mix} - \rho_{air})gH_e}{1 + \left( \frac{\rho_{air}}{\rho_{mix}} \right)\left( \frac{A_o}{A_i} \right)^{\frac{1}{n}}} \right] \right\}^n \tag{4}$$

$$t_f = t_0 - \frac{A_f}{A_o C_d C_u (1-n)} \left[ \frac{1 + \left( \frac{\rho_{air}}{\rho_{mix}} \right)\left( \frac{A_o}{A_i} \right)^{\frac{1}{n}}}{2\left( 1 - \frac{\rho_{air}}{\rho_{mix}} \right)g} \right]^n \left( H_e^{1-n} - H_{max}^{1-n} \right) \tag{5}$$

All values except $H_e$ in Equation (5) are determined by the enclosure integrity test (EIT) and inert agent design [1,2,8,9]. Therefore, when species diffusion is not included, the retention time is determined by the interface height of the air flowing from the outside and the agent–air gas mixture

inside. In other words, when expanding the theoretical equation with *the wide descending interface model*, the height of the interface between air and air–agent mixture gas is adjusted.

### 2.3. Consideration of the Theoretical Models

The theoretical model with and without transport by species diffusion is compared in Figure 3. This figure shows the profile of the agent concentration for the height of the enclosure with and without $O_2$ species diffusion. As previously indicated, *the sharp descending interface model* in Figure 3a does not reflect diffusion. Therefore, the total flux $J_{tot}$ can be expressed as $J_{tot} = J_a$. Based on the leading edge of the interface $H_i$ between air and air–agent mixture gas, the upper part is filled with air ($\rho_{air}$, agent concentration $c = 0$) and the lower part is interpreted to be filled with the initial air–agent mixture gas ($\rho_{mix}$, agent concentration $c = c_0$).

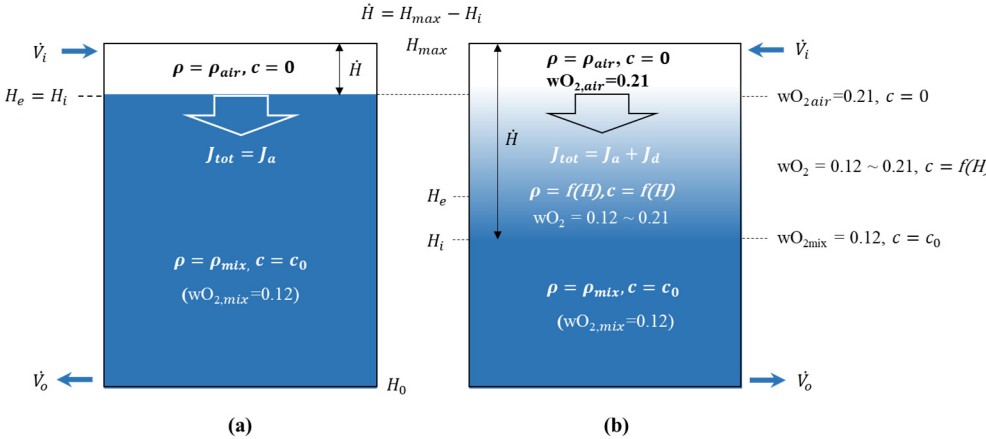

**Figure 3.** Schematic diagram of the z-direction diffusion flux with and without species diffusion: (**a**) Without species diffusion; (**b**) with species diffusion.

However, Figure 3b reflects the species diffusion, and the total flux $J_{tot}$ can be expressed as $J_{tot} = J_a + J_d$. This is because when reflecting species diffusion, the diffusion flux $J_d$ is no longer zero. At this time, diffusion occurs between the air and the air–agent mixture gas due to the difference in concentration between each species [18]. Therefore, the interface has a thickness, and the concentration and density of the agent at the interface decrease as a function of the height. This is a combination of the advection–diffusion equation for the descending interface model with species diffusion. The equation expresses the physical phenomena of how particles, energy, or other physical quantities are transferred inside a physical system due to the two processes of diffusion and advection [19].

In the existing theoretical equation, the retention time is calculated as the interface descending velocity $\partial \dot{H}_i / \partial t$, as shown in Equation (3). The interface descending velocity $\partial \dot{H}_i / \partial t$ reflects the advection velocity $v_a$, but not the diffusion velocity $v_d$. Therefore, if species diffusion is reflected in the interface descending velocity $\partial \dot{H}_i / \partial t$, it can be expressed as follows:

$$\frac{\partial \dot{H}_i}{\partial t} = -(v_a + v_d) = -\left( \frac{\dot{V}_o}{A_f} + v_d \right) \tag{6}$$

The purpose of this study is to calculate the interface descending velocity reflecting the diffusion velocity $v_d$ of the inert agent through experiments and numerical analyses. Therefore, this study proposes a method to quantify the leading edge of the interface $H_i$ by measuring the interface slope $\alpha$ through an experimental study. In addition, a method for calculating the retention time including species diffusion by measuring diffusion velocity $v_d$ through experiments and numerical analysis is also proposed. In addition, the retention time calculated using the newly proposed method is compared

and verified with the measured retention time, and the difference with the existing theoretical models is compared.

### 2.4. Definition of the Interface Slope $\alpha$

In this study, a method to calculate the theoretical retention time is proposed using the interface slope $\alpha$ measured by experiments and numerical analysis. Therefore, it is necessary to define the interface slope $\alpha$. The definition of the interface slope $\alpha$ can be seen in Figure 4.

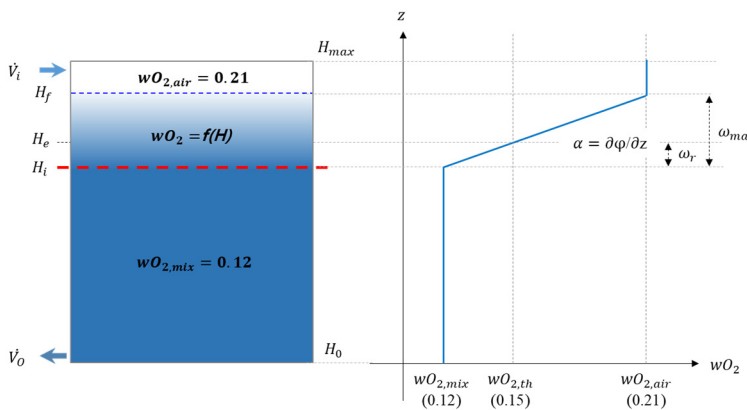

**Figure 4.** Schematic diagram for the $O_2$ concentration distribution profile for calculating interface slope $\alpha$.

This figure shows the $O_2$ concentration profile for the enclosure height. Here, the interface slope $\alpha$ is defined as follows:

$$\alpha = \frac{\partial wO_2}{\partial z} \tag{7}$$

That is, the magnitude of the interface slope $\alpha$ is proportional to the magnitude of the z-direction diffusion flux $J_d$, which is Fick's first law [18].

$$J_d = -D_{ij}\frac{\partial \varphi_{o_2}}{\partial z} \propto \alpha \tag{8}$$

$$D_{ij} = 0.0018583 \frac{T^{\frac{3}{2}}}{p\left(\sigma_{ij}\right)^2 \Omega_{ij}} \sqrt{\left(\frac{1}{M_i} + \frac{1}{M_j}\right)} \left[cm^2/s\right] \tag{9}$$

where

$$\sigma_{ij} = \frac{1}{2}\left(\sigma_i + \sigma_j\right), \ \Omega_{ij} = f\left(\frac{k_B T}{\varepsilon_{ij}}\right), \ \frac{\varepsilon_{ij}}{k_B} = \left(\frac{\varepsilon_i}{k_B}\frac{\varepsilon_j}{k_B}\right)^{\frac{1}{2}}$$

That is, interface slope $\alpha$ means z-direction diffusion flux $J_d$ at the end, and comparison of the interface slope $\alpha$ for each inert agent means comparison of the z-direction species diffusion.

Figure 4 shows the indication of $O_2$ concentration for enclosure height. Here, the *y*-axis is represented by the enclosure height (independent variable) and the *x*-axis is represented by the $O_2$ concentration (dependent variable). This is for comparison with the same axis as Figure 4, which shows the cross-section of the enclosure. This expression method has already been used in several studies [1–4,9,12]. Therefore, this is applied equally to all graphs representing the experimental and numerical analysis results of this study.

## 3. Methodology

### 3.1. Experiment Study

All experiments were conducted to validate the species diffusion for an inert agent. The enclosure used in the experiment was 1 m × 1 m × 2 m in height. A schematic diagram of the enclosure is shown in Figure 5.

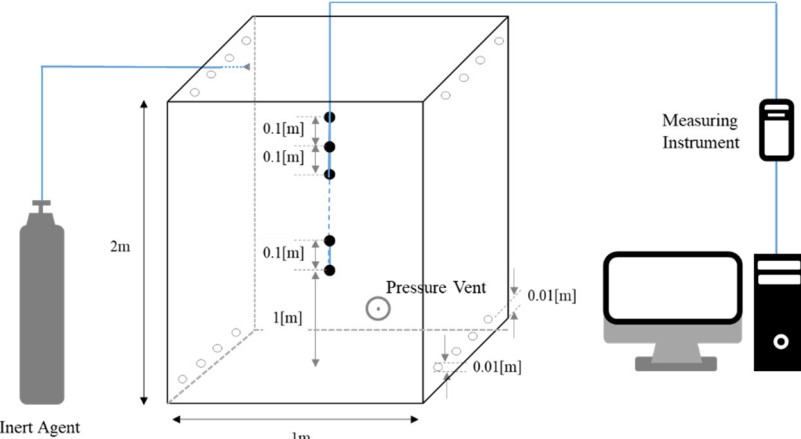

**Figure 5.** Schematic diagram of test enclosure and measuring equipment.

A hole with a diameter of 0.01 m was drilled at an offset of 0.01 m from the floor and the ceiling. The low leakage factor F was set to 0.5, which is the worst condition. That is, the leakage areas at the top and bottom were set identically. The worst condition of F is defined in NFPA 2001 [6]. In order to control the total flow rate of the inflow and outflow, holes were removed or reopened using a vacuum adhesive. The enclosure was placed inside a room with temperature and humidity control, which allowed the external ambient conditions to be controlled. A sensor (OXY-GEN oxygen monitor, AΩ) was placed inside the enclosure to measure the ambient temperature, humidity, and $O_2$ concentration. In this experiment, the $O_2$ concentration for 10 points was measured at 0.1 m intervals from 1 to 1.9 m from the bottom to analyze the interface descending velocity $\partial \dot{H}_i / \partial t$ and interface characteristics. The $O_2$ concentration at each point was measured at 5 s intervals.

The difference between the outside and inside temperatures of the enclosure was within 1 K. This was done to minimize the effect of temperature. However, it was not possible to control the humidity inside the enclosure. Therefore, the humidity prior to the injection of the inert agent to the enclosure was set to a constant so that the humidity under experimental conditions could be matched as much as possible. Three types of inert agents were used. The experimental initial conditions inside and outside the enclosure are listed in Table 2.

**Table 2.** Initial conditions inside and outside the enclosure.

| Case | Total Leakage Area (cm$^2$) | Inert Agent | Temperature In/Out (K) | Initial Humidity In/Out (%) | Inside Pressure (hPa) |
|---|---|---|---|---|---|
| 1-1 | 6.28 (Hole 8ea) | IG-01 | 292.5/293.4 | 20/50 | 1014 |
| 1-2 | | IG-55 | 293.3/293.2 | 21/51 | 1014 |
| 1-3 | | IG-541 | 292.8/293.3 | 19/50 | 1013 |
| 2-1 | 9.42 (Hole 12ea) | IG-01 | 292.7/293.4 | 21/50 | 1014 |
| 2-2 | | IG-55 | 293.3/293.3 | 20/51 | 1014 |
| 2-3 | | IG-541 | 293.1/293.4 | 19/51 | 1013 |

The experiment was repeated 5 times for each case, and the measured data were used to calculate the interface slope $\alpha$ and diffusion velocity $v_d$. As in previous studies, agent discharge was intentionally omitted [3,4,15,16]. In this study, the design $O_2$ concentration in the air–agent mixture gas was set to 0.12, and the $O_2$ concentration that determines the retention time was set to 0.15. The inert agent was injected into the enclosure at a pressure of 1.2 MPa to thoroughly mix the air and agent. Therefore, the initial $O_2$ concentration according to the height of the measurement could be set to $0.12 \pm 1 \times 10^{-3}$.

### 3.2. Numerical Study

Numerical analyses were additionally performed to obtain specific information that is difficult to interpret in the experiment. The domain used in the numerical modeling was the same size as the enclosure of the actual experiment. Domain 1 is a square box with a capacity of 1 m × 1 m × 2 m height and is the same size as the enclosure used in the experiment. Domain 2 is the external space and is a rectangular box with a capacity of 5 m × 5 m × 3 m height, which is approximately 37.5 times larger than Domain 1. This was used to predict the incoming and outcoming fluid flow more accurately. The numerical modeling was performed in three dimensions, and the computational domains are shown in Figure 6.

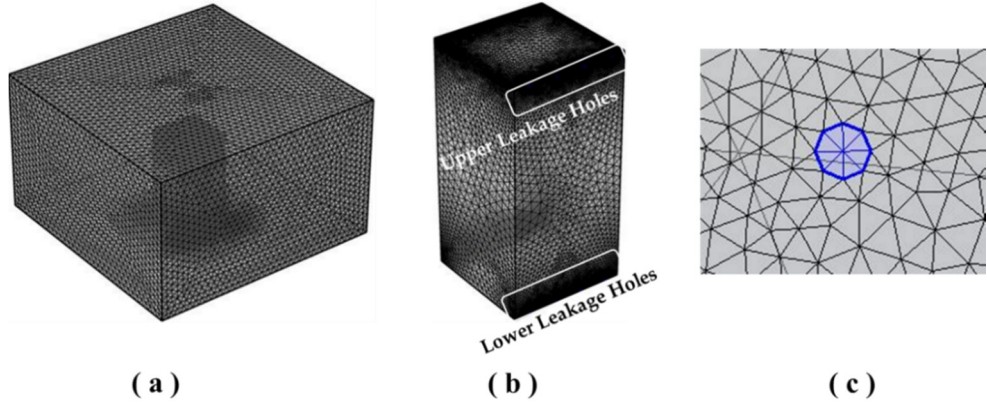

**Figure 6.** Overview of the geometry and mesh used in numerical analysis: (**a**) Overview of the total domain; (**b**) overview of the enclosure; (**c**) overview of the leakage hole.

The computational fluid dynamics (CFD) tool used in this numerical analysis was COMSOL Multiphysics (Version 5.5, 2019, Altsoft). The Reynolds-Averaged Navier-Stokes (RANS) turbulence model type was applied with a $k$–$\mathcal{E}$ turbulence model based on the analysis of two additional partial differential equations for the kinetic energy, $k$, and the energy dissipation rate, $\mathcal{E}$.

In addition, a mesh-sensitive study was conducted using the results of the change in the $O_2$ concentration over time of each agent measured in the experiment. The mesh conditions used in the mesh-sensitive study are shown in Table 3 and Figure 7. The initial boundary conditions of the numerical analysis to match the results of the experiment are listed in Table 4.

**Table 3.** List of mesh sizes and elements for modeling.

| Test | Outside (Domain 2) | Inside (Domain 1) | Leakage Area | Domain Elements |
|------|--------------------|--------------------|--------------|-----------------|
| 1 | 0.197–0.564 [m] | 0.07–0.18 [m] | 0.005–0.009 [m] | 1,177,357 |
| 2 | 0.134–0.371 [m] | 0.06–0.15 [m] | 0.003–0.007 [m] | 1,809,238 |
| 3 | 0.134–0.371 [m] | 0.04–0.12 [m] | 0.002–0.005 [m] | 2,500,857 |

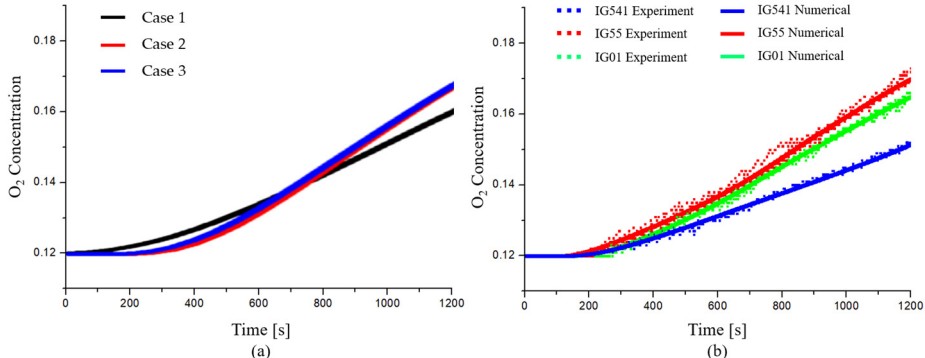

**Figure 7.** Mesh-sensitive study of the $O_2$ concentration: (**a**) Change in the $O_2$ concentration according to mesh sizes and elements; (**b**) comparison of the $O_2$ concentration between experimental and numerical results at the specified height $H_e$

**Table 4.** List of initial conditions inside and outside the enclosure taken for modeling.

| Case | Total Leakage Area (cm$^2$) | Inert Agent | Density (kg/m$^3$) | Initial $O_2$ Concentration | Temperature Inside (K) | Temperature Outside (K) |
|------|------|------|------|------|------|------|
| 1-1 | 5.79 (Hole 8ea) | IG-01 | 1.3988 | 0.120 | 290.5 | |
| 1-2 | | IG-55 | 1.2920 | 0.119 | 290.5 | 293 |
| 1-3 | | IG-541 | 1.2978 | 0.120 | 289.5 | |
| 2-1 | 8.68 (Hole 12ea) | IG-01 | 1.3988 | 0.120 | 290.5 | |
| 2-2 | | IG-55 | 1.2920 | 0.119 | 290.5 | 293 |
| 2-3 | | IG-541 | 1.2978 | 0.119 | 289.5 | |

The mesh was configured by separating Domain 1, Domain 2, and the leakage areas. In the mesh-sensitive study, the mesh condition showed that the $O_2$ concentration was saturated in test 2. This·can be confirmed through Figure 7. Therefore, the mesh consists of 1,809,238 domain elements, 40,042 boundary elements, 1082 edge elements, and a 1.08 maximum element growth rate. Other numerical conditions such as dynamic viscosity ($\eta_{mix}$), thermal conductivity (k, Pr = 0.7), and heat capacity ($C_{p,mix}$) were performed using COMSOL Multiphysics 5.5 [20–23].

## 4. Results

### 4.1. Experiment Analysis

First, the experimental $O_2$ concentration data were measured by the height of the enclosure in order to obtain the interface slope $\alpha$ and interface descending velocity $\partial \dot{H}_i / \partial t$. Therefore, $O_2$ concentration was measured for a total of 10 points at 0.1 m intervals from 1 to 1.9 m in height of the enclosure. The $O_2$ concentration data according to the measured height are shown in Figure 8. As explained in Chapter 2, Figure 8 shows the *y*-axis as the enclosure height (independent variable) and the *x*-axis as the $O_2$ concentration (dependent variable) to express the $O_2$ concentration data measured at the specified height.

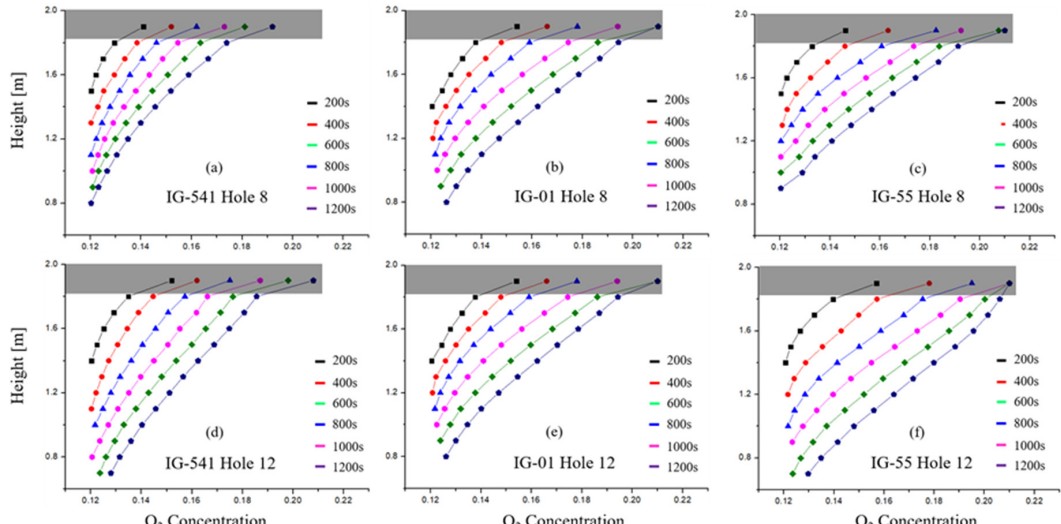

**Figure 8.** Change in the interface slope $\alpha$ over time for each agent (average value of 5 data).

Figure 8a–c present the conditions with 8 leakage holes, and Figure 8d–f present the conditions with 12 leakage holes. The $O_2$ concentration measurement with an increase in the leakage hole was measured to observe how species diffusion (diffusion flux $J_a$) changes with an increase in outflow volumetric flow $\dot{V}_o$.

Figure 8 shows how much $O_2$ introduced from the outside diffuses into the z-direction over time. In this figure, it can be seen that the interface slope $\alpha$ saturates after a certain period of time (approximately 200 s). In other words, it means that the $O_2$ introduced from the outside diffuse through the z-direction with a constant diffusion flux $J_d$ (i.e., diffusion velocity $v_d$).

The interface slope $\alpha$ appeared to differ depending on the type of inert agent used in the experiment. The interface slope $\alpha_{541}$ of IG-541 was the smallest and $\alpha_{55}$ of IG-55 was the largest. It can be seen that this is the same as the results of a previous study [3], that is, because the interface slope $\alpha$ represents the z-direction diffusion flux $J_d$, it was confirmed that the $J_d$ of each inert agent presents different values.

In addition, the interface slope $\alpha$, that is, the diffusion flux $J_d$, does not change significantly with the outflow volumetric flow $\dot{V}_o$. This means that the diffusion flux $J_d$ is an independent factor that is not affected by advection flux $J_a$. Moreover, because the theoretical models for the retention time do not consider changes in temperature, the binary diffusion coefficient $D_{ij}$ can be seen as a fixed value [24,25]. This means that by Fick's first law, the diffusion flux $J_d$ can also be seen as a fixed value. Therefore, in the theoretical equation for retention time, the diffusion flux $J_d$ can be calculated as a constant.

The $O_2$ concentration near the ceiling of the enclosure showed a tendency to increase rapidly. This is also the same as the results of a previous study [3]. This can be inferred from the turbulent mixing formed by the inflow volumetric flow $\dot{V}_i$. Therefore, this part will be analyzed in detail through numerical analysis. Therefore, the analysis of the interface slope $\alpha$ of this part was excluded.

Figure 9 shows the interface slope $\alpha$ and the maximum interface thickness $\omega$ for each inert agent. That is, this figure shows how, when the amount of $O_2$ introduced from the outside is changed, the interface slope $\alpha$ and the interface thickness $\omega$ change when the $O_2$ concentration threshold value is reached at the specified height $H_e$ and ceiling of the enclosure $H_{max}$. Therefore, the theoretical equation of the newly proposed model can be completed in Figure 9.

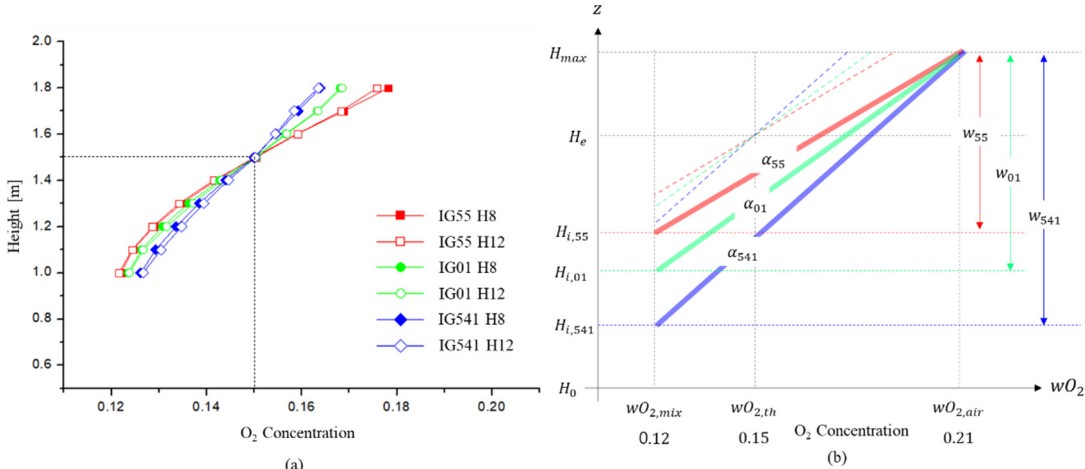

**Figure 9.** Comparison of the interface slope $\alpha$ according to the number of leakage holes of each agent: (**a**) Comparison of the interface slope $\alpha$ for each agent when the $O_2$ concentration at the specified height $H_e$ is 0.15; (**b**) comparison of the maximum interface thickness $\omega_{max}$ of each agent when the $O_2$ concentration at the maximum height $H_{max}$ of the enclosure is the $O_2$ concentration of air.

Figure 9a shows a comparison of the measured $O_2$ concentration for each height of the enclosure for each agent when the specified height $H_e$ (1.5 m) has an $O_2$ concentration of 0.15. The $O_2$ concentration value is the average value of the data measured 5 times. In this figure, the difference in the interface slope $\alpha$ according to each agent is shown, and it can be seen that the interface slope $\alpha$ (i.e., diffusion flux $J_d$) is not significantly affected by the outflow volumetric flow $\dot{V}_o$. The interface slope $\alpha$ of each agent was measured as $\alpha_{541} = 0.062$, $\alpha_{01} = 0.071$, and $\alpha_{55} = 0.083$.

Figure 9b shows the maximum interface thickness $\omega_{max}$ and the leading edge of the interface $H_i$ for each agent. From this figure, it can be seen that if the interface slope $\alpha$ is known, the interface thickness $\omega$ and the leading edge of the interface $H_i$ can be obtained.

The relationship between the interface slope $\alpha$ and the maximum interface thickness $\omega_{max}$ can be inferred by Figure 9b as follows:

$$\alpha = \frac{\partial wO_2}{\partial z} = \frac{(wO_{2,air} - wO_{2,mix})}{\omega_{max}} \tag{10}$$

$$\omega_{max} = \frac{(wO_{2,air} - wO_{2,mix})}{\alpha} \tag{11}$$

The leading edge of the interface $H_i$ can also be calculated using the above equation. First, if the thickness from the specified height $H_e$ to $H_i$ is $\omega_r$, $H_i$ can be expressed as follows:

$$\alpha = \frac{\partial wO_2}{\partial z} = \frac{\left(wO_{2,th} - wO_{2,mix}\right)}{\omega_r} = \frac{\left(wO_{2,th} - wO_{2,mix}\right)}{H_e - H_i} \tag{12}$$

$$H_i = H_e - \frac{\left(wO_{2,th} - wO_{2,mix}\right)}{\alpha} \tag{13}$$

As shown in Equation (13), if the specified height $H_e$ and the $O_2$ threshold concentration $wO_{2,th}$ are determined to measure the retention time, $H_i$ would have a fixed value. Therefore, if the interface slope $\alpha$ and the z-direction velocity $v_d$ can be inferred, the retention time including the species diffusion can be obtained using Equation (14).

$$\int_{H_{max}}^{H_i} \partial \dot{H}_i = -\int_{t_0}^{t_r} (v_a + v_d)\partial t = -\int_{t_0}^{t_f} \left(\frac{\dot{V}_o}{A_f} + v_d\right)\partial t \tag{14}$$

To obtain the retention time using Equation (14), it is necessary to know the information about the remaining factors. Here, $H_{max}$ and $A_f$ are design factors, and the outflow volumetric flow $\dot{V}_o$ can be obtained through calculation or EIT. Therefore, if the interface slope $\alpha$ and diffusion velocity $v_d$ are known, a theoretical calculation of the retention time is possible.

The interface descending velocity $\partial \dot{H}_i / \partial t$ can be calculated using the measured $O_2$ concentration data according to the enclosure height. This can be seen in Figure 10.

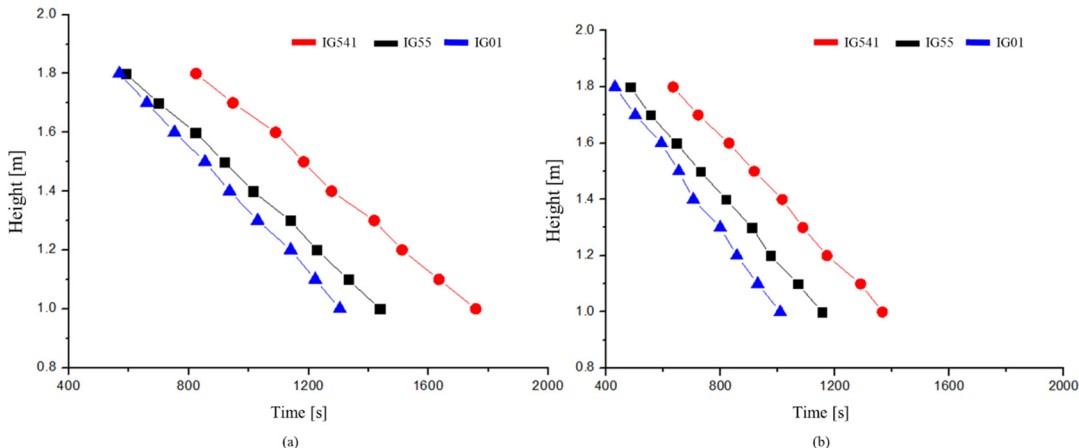

**Figure 10.** Comparison of the experimental results of the interface descending velocity $\partial \dot{H}_i / \partial t$ when the $O_2$ concentrations were 0.15: (**a**) Comparison of 8 holes; (**b**) comparison of 12 holes.

Figure 10 shows the time to reach an $O_2$ concentration of 0.15 per 0.1 m from 1.8 to 1 m of enclosure height for the IG-55, IG-01, and IG-541. In other words, the slope of this figure shows the interface descending velocity $\partial \dot{H}_i / \partial t$.

From this figure, it can be seen that the interface descending velocity $\partial \dot{H}_i / \partial t$ descends in the z-direction at a constant velocity. This shows the same result regardless of the type of inert agent used. However, the interface descending velocity $\partial \dot{H}_i / \partial t$ is different for each agent. This can be attributed to the difference between the outflow volumetric flow $\dot{V}_o$ and diffusion flux $J_d$. The outflow volume flow $\dot{V}_o$ is expected to be similar because the densities of IG-55 and IG-541 are almost the same. Therefore, the advection velocity $v_a$ is expected to be similar according to Equation (3). However, as the interface slope $\alpha$ is larger for IG-55 than for IG-541, the diffusion velocity $v_d$ is expected to be larger for IG-55. Therefore, the interface descending velocity $\partial \dot{H}_i / \partial t$ for IG-55 is greater than that for IG-541. In addition, because IG-01 has a higher density than IG-55, the advection velocity $v_a$ will be greater for IG-01. However, because the diffusion velocity $v_d$ of IG-55 is large, the interface descending velocities $\partial \dot{H}_i / \partial t$ for IG-01 and IG-55 show small differences. As shown in Figure 10b, this tendency is also the same when the outflow volume flow $\dot{V}_o$ increases.

The retention time for each agent was measured. It was measured at a specified height of 1.5 m, based on an $O_2$ concentration of 0.15. This is shown in Figure 11.

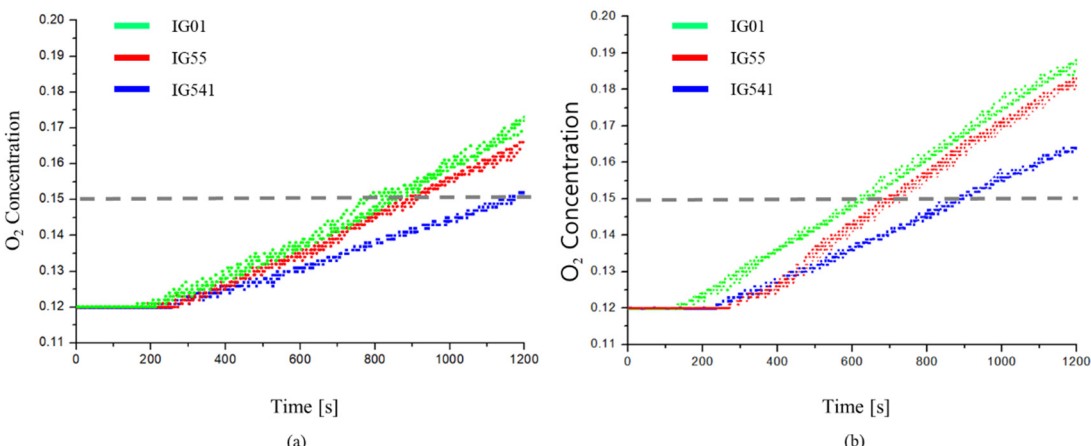

**Figure 11.** Change in the $O_2$ concentration over time for each inert agent: (**a**) 8 holes; (**b**) 12 holes.

Figure 11 shows a comparison of $O_2$ concentrations for the three agents with 8 and 12 leakage areas. In Figure 11, the retention time is the fastest for IG-01 and the slowest for IG-541. When compared, the retention time of IG-55 was about 260 s faster than that of IG-541. However, IG-55 and IG-541, on the other hands, have almost the same density. That is, there is no difference in outflow volume flow $\dot{V}_o$, i.e., advection flow $J_a$. Therefore, it is determined that the difference in retention time between the two agents is ultimately caused by the difference in diffusion flux $J_d$. As IG-55 has a larger $J_d$ than IG-541, even though $J_a$ is the same, the retention time appears faster.

Moreover, there is a minor difference between the retention times of IG-01 and IG-55. The diffusion flux $J_d$ is greater for IG55 than for IG-01, but the retention time is faster for IG-01. This can be explained by the difference in the advection flux $J_a$ due to the difference in density between the agents. The density of IG-01 is 1.399 kg/m$^3$, which is approximately 8.3% greater than the density of IG-55, 1.292 kg/m$^3$. As a result, it can be seen from Equation (4) that the outflow volumetric flow $\dot{V}_o$ is larger for IG-01 than for IG-55. Therefore, the difference in retention time between the two agents is reduced, and it is determined to have similar values.

The experimental data measured and the data calculated using the proposed equation are summarized in Table 5. In Table 5, the retention time theoretically calculated using the measured and calculated factors is compared to the measured retention time.

**Table 5.** Summary of experimental results for each inert agent.

| Inert Agent | Inside Density $\rho_{mix}$ (kg/m$^3$) | The Edge of the Interface $H_i$ (m) | Interface Slope $\alpha$ | Interface Descending Velocity $\partial \dot{H}_i / \partial t$ (m/s) | Retention Time H8 Calc./Mea. (s) | Retention Time H12 Calc./Mea. (s) |
|---|---|---|---|---|---|---|
| IG-541 | 1.292 | 1.016 | 0.062 | H8: $-8.32 \times 10^{-4}$<br>H12: $-1.07 \times 10^{-3}$ | 1183/1164 | 918/901 |
| IG-55 | 1.294 | 1.139 | 0.083 | H8: $-9.38 \times 10^{-4}$<br>H12: $-1.21 \times 10^{-3}$ | 918/892 | 711/702 |
| IG-01 | 1.399 | 1.078 | 0.071 | H8: $-1.08 \times 10^{-3}$<br>H12: $-1.41 \times 10^{-3}$ | 854/837 | 654/636 |

$H_i$: The height of the edge of the interface when the $O_2$ concentration of $H_e$ (1.5 m) is 0.15.

The experiment was performed using three types of inert agents. In this experiment, the interface slope $\alpha$ was the smallest with IG-541 ($\alpha_{541} = 0.062$), it was $\alpha_{01} = 0.071$ for IG-01, and it was the largest at $\alpha_{55} = 0.083$ for IG-55. The retention time of IG-541 with the smallest interface slope $\alpha_{541}$ was the longest. The edge of the interface $H_i$ at an $O_2$ concentration of 0.15 of the specified height $H_e$ can be

calculated using the interface slope $\alpha$ (Equation (13)). Therefore, the retention time can be calculated from the interface descending velocity $\partial \dot{H}_i/\partial t$ and the edge of the interface $H_i$ (Equation (14)).

However, based on the aforementioned data, there was a difference between the retention time calculated using Equation (14) and that measured in an actual experiment. The retention time calculated in theory is approximately 3% slower than the retention time measured in the experiment. This part is considered as due to the turbulence mixing zone caused by the inflow volumetric flow $\dot{V}_i$, along with the average error of the calculated factors. Therefore, further analysis of this error is conducted through numerical analysis.

A summary of the experimental results performed so far is as follows. First, it was found that the interface slope $\alpha$, that is, the diffusion flux $J_d$, had a constant size after a certain time and had a unique value that was different for each agent. Second, through the interface slope $\alpha$, the edge of the interface $H_i$, which is the interface between air and air–agent mixture gas, could be calculated. Finally, the retention time reflecting the diffusion velocity $v_d$ yielded similar results to the measured retention time.

However, there was a limit to the information obtained from the experimental results. The difference in interface slope $\alpha$ for each agent could be interpreted as a difference in diffusion flux $J_d$, that is, diffusion velocity $v_d$, but because the outflow volumetric flow $\dot{V}_o$ could not be measured, the diffusion velocity $v_d$ of each inert agent could not be calculated. In addition, it was difficult to obtain information on the cause of the slope transients at the top of the enclosure, which are expected to cause differences in the measured and calculated retention times. Therefore, in the next chapter, more detailed information was obtained through numerical analysis.

### 4.2. Numerical Analysis

The interface slope $\alpha$ for each agent was measured through the experiments and was confirmed to be unaffected by the z-direction advection flux $J_a$. Then, the edge of the interface $H_i$ was calculated by measuring the interface slope $\alpha$ for each agent. In addition, by measuring the interface descending velocity $\partial \dot{H}_i/\partial t$, it was also possible to confirm the difference between the calculated and measured retention times.

However, it was challenging to obtain all the information regarding the retention time of the inert agent through experiments. Information regarding the outflow volumetric flow $\dot{V}_o$ and turbulence mixing zone of each agent was insufficient. Therefore, numerical analysis was performed to obtain additional information that was lacking in the experiment.

First, Figure 12 presents the retention time at a specified height $H_e$ of 1.5 m and the outflow volumetric flow $\dot{V}_o$ for each agent. The comparison between the experiment and numerical analysis for the retention time was optimized through the mesh-sensitive test in the previous chapter.

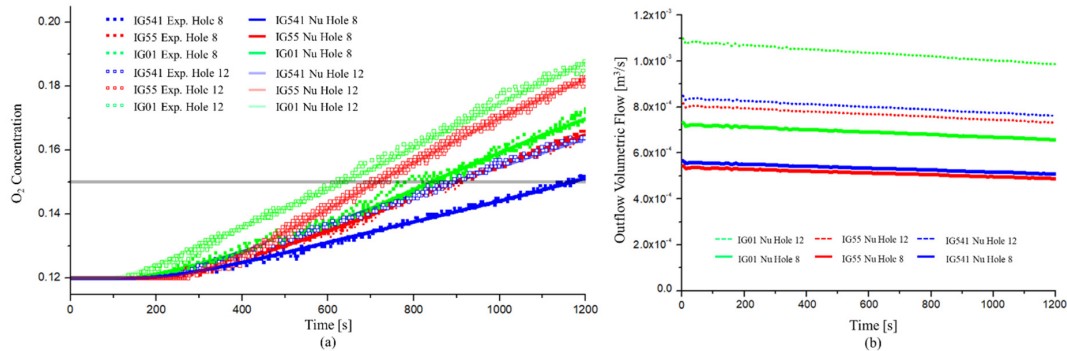

**Figure 12.** Comparison of experimental and numerical results for the retention time of each agent: (**a**) $O_2$ concentration over time; (**b**) outflow volumetric flow $\dot{V}_o$ over time.

Figure 12a shows the numerical and experimental results of the $O_2$ concentration over time. In Figure 12a, the numerical results of the $O_2$ concentration over time also presented the same results as those of the experiment. For the 8 holes, the retention time of IG-01 was the fastest and that of IG-541 was the slowest. For 12 holes, the results of the $O_2$ concentration over time also yielded similar results.

Figure 12b shows the results for the outflow volumetric flow $\dot{V}_o$. As the outflow volumetric flow $\dot{V}_o$ is proportional to the density of the air–agent mixture gas, IG-01 showed the largest value. In addition, the increase in leakage holes resulted in an increase in the outflow volumetric flow $\dot{V}_o$. It can be seen that this is the same result as the theory in the previous chapter.

In order to define the newly proposed theoretical equation for the retention time, it is necessary to determine the diffusion velocity $v_d$. The interface descending velocity $\partial \dot{H}_i/\partial t$ was measured through experiments; thus, the diffusion velocity $v_d$ can be calculated using Equation (6) if the advection velocity $v_a$ is known. Therefore, the advection velocity $v_a$ can be calculated using the outflow volumetric flow $\dot{V}_o$ measured through numerical analysis. The outflow volumetric flow $\dot{V}_o$ and advection velocity $v_a$ are listed in Table 6.

**Table 6.** Comparison of the z-direction advection velocity $v_a$ and the z-direction diffusion velocity $v_d$ for each inert agent.

| Inert Agent | Inside Density $\rho_{mix}$ (kg/m$^3$) | Descending interface Velocity $\partial \dot{H}_i/\partial t$ (m/s) | | Averaged Advection Velocity $v_a$ (m/s) | | Diffusion Velocity $v_d$ (m/s) | |
|---|---|---|---|---|---|---|---|
| | | 8 Holes | 12 Holes | 8 Holes | 12 Holes | 8 Holes | 12 Holes |
| IG-541 | 1.292 | $-8.32 \times 10^{-4}$ | $-1.07 \times 10^{-3}$ | $-4.92 \times 10^{-4}$ | $-7.32 \times 10^{-4}$ | $-3.40 \times 10^{-4}$ | $-3.41 \times 10^{-4}$ |
| IG-55 | 1.294 | $-9.38 \times 10^{-4}$ | $-1.21 \times 10^{-3}$ | $-5.02 \times 10^{-4}$ | $-7.68 \times 10^{-4}$ | $-4.36 \times 10^{-4}$ | $-4.43 \times 10^{-4}$ |
| IG-01 | 1.399 | $-1.08 \times 10^{-3}$ | $-1.41 \times 10^{-3}$ | $-6.68 \times 10^{-4}$ | $-9.87 \times 10^{-4}$ | $-4.12 \times 10^{-4}$ | $-4.23 \times 10^{-4}$ |

Table 6 indicates that the z-direction advection velocity $v_a$ is proportional to $\rho_{mix}$ because it is a function of the outflow volumetric flow $\dot{V}_o$, and the z-direction diffusion velocity $v_d$ is proportional to the interface slope $\alpha$ (i.e., diffusion flux $J_d$). Therefore, it was confirmed that the diffusion velocity $v_d$ of each agent was not affected by the outflow volumetric flow $\dot{V}_o$.

In addition, species diffusion is determined by the difference in species concentration and the binary diffusion coefficient $D_{ij}$, as can be seen from Fick's law [18]. As all of the theoretical models for the retention time, including this study, do not consider the effect of temperature, the binary diffusion coefficient $D_{ij}$ of each agent can also be treated as a constant value [24,25]. That is, the interface slope $\alpha$ and the binary diffusion coefficient $D_{ij}$ can be kept constant regardless of the outflow volumetric flow $\dot{V}_o$ (i.e., advection flux $J_a$) and the geometry of the enclosure. Therefore, when the retention time is calculated using the theoretical equation, the z-direction diffusion velocity $v_d$ for each inert agent can be calculated by substituting a fixed value. From Table 6, it can be seen that the difference in species diffusion for each inert agent occurs clearly.

Based on the optimized numerical analysis results, the turbulent mixing zone of the upper part of the enclosure that could not be analyzed in the experiment was analyzed. Experiments have shown that this turbulent mixing region can cause distortion in the interface slope $\alpha$ and the theoretical retention time. As mentioned in the experimental study, a turbulence mixing zone is created near the ceiling of the enclosure due to incoming air. Therefore, turbulent mixing occurs between air and air–agent mixture gas in this zone by the driving force of the inlet volume flow $\dot{V}_i$. This is shown in detail in Figure 13.

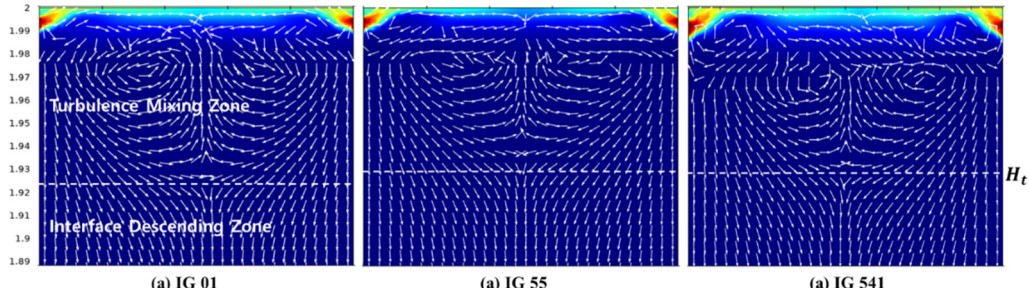

**Figure 13.** Schematic diagram of turbulence mixing zone and interface descending zone.

Figure 13 presents the cross-sectional area of the enclosure for each inert agent at $t = 10$ s. The arrow in the figure represents the species flow field inside the enclosure; air introduced from the outside in the turbulence mixing zone is mixed in all directions by advection. Therefore, it is not possible to interpret this part simply by the z-direction descending interface. Rather, this area can be considered as if mixing occurs in all areas of the turbulence mixing zone, similar to *the continuous mixing model*. Therefore, the turbulent mixing area may be interpreted as a region having a single concentration rather than a concentration gradient in the z-direction. It can be interpreted that the starting point of the interface is not $H_{max}$ when calculating the retention time, but $H_t$, which is the boundary between the turbulent mixing region and the interface descending region. In addition, it can be expected that $t_0$, the starting time of the retention time measurement, should be approximately 10 s instead of 0 s. Therefore, it can be seen that the distortion of the interface slope $\alpha$ and the theoretical retention time occur in this zone. In addition, the error between the calculated retention time and the measured retention time can be seen as being caused by this zone.

### 4.3. Summary of the Experimental and Numerical Analysis Results

Through experiments and numerical analysis, information regarding the interface slope $\alpha$, interface thickness $\omega$, leading edge of the interface $H_i$, turbulence mixing zone, and z-direction diffusion velocity $v_d$ was obtained. Therefore, the results of this study can be expressed in Figure 14 and Table 7. Table 7 summarizes the results obtained through experiments and numerical analysis, and Figure 14 shows the newly proposed model as a figure based on the experiment and numerical analyses.

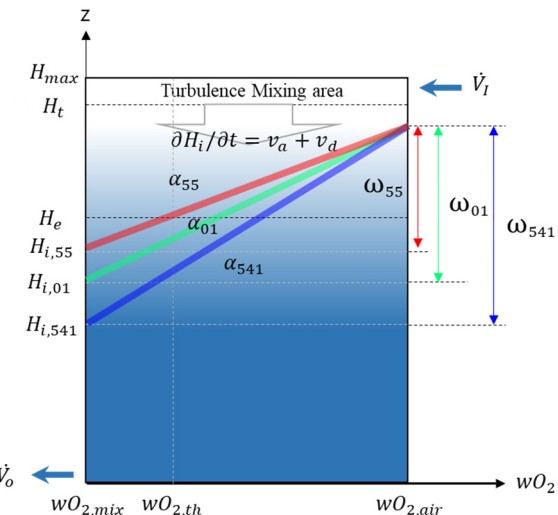

**Figure 14.** Schematic diagram of $O_2$ concentration distribution profile reflecting the experimental and numerical analysis results.

**Table 7.** Summary of experimental and numerical results.

| Inert Agents | Interface Slope $\alpha$ (kg/m$^4$) | Diffusion Velocity $v_d$ (m/s) | Measured Retention Time $t_r$ (s) H8/H12 | Calculated Retention Time $t_r$ (s) H8/H12 | Difference (%) Measured /Calculated Retention Time |
|---|---|---|---|---|---|
| IG-541 | 0.062 | $-3.40 \times 10^{-4}$ | 1164/901 | 1183/922 | −1.6%/−2.3% |
| IG-55 | 0.081 | $-4.40 \times 10^{-4}$ | 892/722 | 918/740 | −2.8%/−2.4% |
| IG-01 | 0.071 | $-4.18 \times 10^{-4}$ | 837/636 | 854/654 | −2.0%/−2.75% |

First, the interface slope $\alpha$ has a unique constant value for each agent. As presented in Equation (8), this value is eventually proportional to the z-direction diffusion flux $J_d$. Therefore, the greater the interface slope $\alpha$, the faster the O$_2$ diffusion and the faster the retention time. The interface slope $\alpha_{55}$ of IG-55 is the largest with $\alpha_{55} = 0.083$, and it has values of $\alpha_{01} = 0.071$ and $\alpha_{541} = 0.062$. The interface slope $\alpha$ is ultimately determined by the concentration difference between air and the agent–air gas mixture; thus, it will have a constant value regardless of the enclosure height. Therefore, if the composition of the agent–air gas mixture does not change, the interface slope $\alpha$ has a fixed value.

Second, using the interface slope $\alpha$ and the specified height $H_e$, the leading edge of the interface $H_i$ can be calculated using Equation (12). Therefore, if the leading edge of the interface $H_i$ can be calculated, the retention time considering species diffusion can also be calculated.

Third, the diffusion velocity $v_d$ can also be taken as a fixed value. This is because the interface slope $\alpha$ is proportional to the diffusion flux $J_d$, and the diffusion velocity $v_d$ is also proportional to the diffusion flux $J_d$. Therefore, if temperature is not considered, $v_d$ can also be taken as a fixed value when calculating the theoretical retention time.

Finally, it was confirmed that there was a turbulence mixing zone at the top of the enclosure; it cannot be interpreted as a z-direction descending interface model, because this area induces mixing of the air and the agent–air gas mixture by advection of the air introduced from the outside. Therefore, when calculating the retention time, it was confirmed that more accurate results were obtained when the turbulence mixing zone was excluded.

Based on the results thus far, Table 7 presents the calculated retention time using the theoretical equation proposed in the previous chapter.

The diffusion velocity was $v_{d,541} = -3.40 \times 10^{-4}$, $v_{d,55} = -4.40 \times 10^{-4}$, and $v_{d,01} = -4.18 \times 10^{-4}$. These results were the same as those for the interface slope $\alpha$. The measured and calculated retention times for all the inert agents present errors within 2.8%. This part is considered to be due to the turbulence mixing zone and the average error of the calculated factors.

However, the retention time considering species diffusion proposed in this study is significantly more accurate than the retention time of the existing models. This can be seen in more detail in Figure 15. That is, this figure compares the retention time according to the height measured in the experiment with the retention time of the existing theoretical models, including the theoretical method proposed in this study. A comparison of the existing theoretical model was conducted in *the sharp descending interface model* and *the wide descending interface model*. From this figure, it can be seen that the calculation of the retention time through the newly proposed model can express the actual measured retention time better than the existing models.

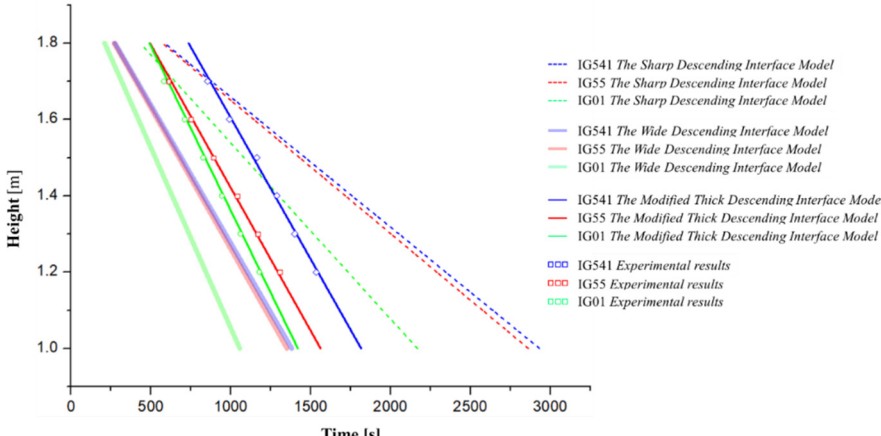

**Figure 15.** Comparison for the retention time according to the height of each model.

First, as *the sharp descending interface model* does not consider species diffusion, the retention time is largely calculated. Therefore, it can be confirmed that this model shows optimistic results for the retention time. In addition, *the wide descending interface model* quickly calculates the species diffusion. Therefore, the retention time was calculated to be faster than the actual measured data. That is, it is confirmed that the results are very conservative.

In addition, *the sharp descending interface model* and *the wide descending interface model* do not reflect the unique species diffusion characteristics (the interface slope $\alpha$ and the diffusion velocity $v_d$) of each inert agent, because the difference between IG-55 and IG-541 is not distinguished.

On the other hand, it was confirmed that the newly proposed model reflects the fixed interface slope $\alpha$ and the z-direction diffusion velocity $v_d$ of each inert agent, thus representing the most accurate retention time. Therefore, when predicting the retention time of the inert agent, if the interface slope $\alpha$ and diffusion velocity $v_d$ proposed in this study can be reflected, a more accurate retention time can be predicted.

## 5. Discussion and Future Work

This study was initiated to demonstrate that species diffusion is necessary to calculate the retention time of inert agents. Therefore, a new theoretical governing equation was proposed to enable a more accurate estimation of retention time. However, in this study, the study was limited to inert agents that are greatly affected by $O_2$ diffusion. Therefore, further studies that can separate halocarbon-compounds and inert agents from FES-gaseous agents and analyze them are needed.

In addition, as with previous studies, this study began with assumptions regarding difficult interpretations aimed at simplifying the theoretical evaluation. The analysis of the flow field effect due to the discharge or the effect of division and stratification of the agent itself was excluded because the inert agent discharge step was omitted. In addition, because the temperatures inside and outside the enclosure are set identically, the analysis of the z-direction advection flux $J_a$ and the z-direction diffusion flux $J_d$ effects, which change depending on temperature, is also excluded. Therefore, it is necessary to further study the effect of these effects on retention time through future studies.

In addition, it seems necessary to further study the reason the interface slope $\alpha$ proposed in this study varies depending on the inert agent and in order to bring interface slope $\alpha$ and diffusion velocity $v_d$ to fixed values of each inert agent. In addition, to predict the retention time more accurately, more geometry structures and input data must be evaluated. Finally, it was confirmed that the turbulence mixing zone formed by the inflow volumetric flow $\dot{V}_i$ affects the theoretical retention time, but further research is needed on how to reflect this zone when calculating the theoretical retention time. Through these additional studies, a theoretical governing equation for a more accurate retention time can be established.

## 6. Conclusions

In general, when evaluating the retention time of an inert agent, the retention time is evaluated as the time at which the initial inert agent concentration in the enclosure is reduced by 15%. However, in this study, it was evaluated based on the $O_2$ concentration rather than the agent concentration. This is because the inert agent's extinguishing ability is determined by how long the $O_2$ concentration can be maintained below a certain value.

This study proposed a method that can reflect the $O_2$ diffusion when calculating the retention time of an inert agent, and this was verified through experiments and numerical analysis. Through this verification, a fixed interface slope $\alpha$ was proposed for each agent, and the relationship between the interface slope $\alpha$ and the retention time was proved. The interface slope $\alpha$ eventually becomes proportional to the z-direction diffusion flux $J_d$. Therefore, a large interface slope $\alpha$ indicates that the z-direction diffusion flux $J_d$ is large, which means that the retention time is prolonged.

In addition, it was found that the turbulence mixing zone existed at the top of the enclosure owing to the turbulence of the incoming air. It was confirmed that the distortion of the interface slope $\alpha$ was generated by this area. Therefore, it was confirmed that the difference between the theoretical retention time and the measured retention time was eventually caused by the turbulence mixing zone.

When $O_2$ diffusion is reflected in the inert agent, the interface descending velocity $\partial \dot{H}_i / \partial t$ is calculated as the sum of the z-direction advection velocity $v_a$ and the z-direction diffusion velocity $v_d$. Here, it was confirmed that the z-direction advection velocity $v_a$ is proportional to the outflow volumetric flow $\dot{V}_o$, and the z-direction diffusion velocity $v_d$ is proportional to the interface slope $\alpha$. Therefore, assuming that there is no difference in temperature inside and outside the enclosure, it was confirmed that the z-direction diffusion velocity $v_d$ between the inert agents has a fixed value.

Finally, it was confirmed that the theoretical retention time applied to the interface slope $\alpha$, the leading edge of the interface $H_i$, and the diffusion velocity $v_d$ can reduce the difference from the measured retention time to within approximately 3%.

**Author Contributions:** Conceptualization, G.K. and H.-J.L.; data curation, G.K. and J.-H.C.; formal analysis, G.K., J.-H.C. and H.-J.L.; investigation, G.K. and J.-H.J.; software, G.K.; project administration, H.-J.L.; writing—original draft preparation, G.K.; writing—review and editing, H.-J.L. All authors have read and agreed to the published version of the manuscript.

**Funding:** This work was supported by a 2-Year Research Grant of Pusan National University.

**Conflicts of Interest:** The authors declare no conflict of interest.

## Nomenclature

| | |
|---|---|
| $A_f$ | Enclosure floor area [m$^2$] |
| $A_o$ | Total cross-sectional area of the lower leakage area [m$^2$] |
| $C_d$ | Discharge coefficient in the orifice flow equation (dimensionless quantity) |
| $C_o$ | Discharge coefficient for the orifice described by $A_o$ (dimensionless quantity) |
| $C_u$ | Unit conversion constant in the orifice flow equation (various) |
| $D_{ij}$ | Binary diffusion coefficient between molecules $i$th and $j$th [m$^2$/s] |
| $F$ | Lower leakage fraction (dimensionless quantity) |
| $g$ | Acceleration due to gravity (9.81 m/s$^2$) |
| $H_e$ | Interface specified height [m] |
| $H_i$ | Leading edge of the interface [m] |
| $H_{max}$ | Maximum enclosure height [m] |
| $J_a$ | Advection flux [kg/m$^2$/s] |
| $J_d$ | Diffusion flux [kg/m$^2$/s] |
| $J_{tot}$ | Total flux [kg/m$^2$/s] |
| $k_B$ | Boltzmann constant |
| $M$ | Molecular mass [g/mol] |
| $n$ | Orifice flow exponent |

| | |
|---|---|
| p | Pressure [atm] |
| T | Temperature [K] |
| $t_r$ | Retention time [s] |
| $v_a$ | Advection velocity [m/s] |
| $v_d$ | Diffusion velocity [m/s] |
| $\dot{V}_o$ | Outflow volumetric flow [m$^3$/s] |
| $\dot{V}_i$ | Inflow volumetric flow [m$^3$/s] |
| $wO_{2,air}$ | O$_2$ concentration of the air [0.21] |
| $wO_{2,mix}$ | initial O$_2$ concentration of the air-agent gas mixture [0.12] |
| $wO_{2,th}$ | O$_2$ threshold concentration to determine the retention time [under 0.15] |
| $\alpha$ | Interface slope [kg/m$^3$/m] |
| $\omega$ | Interface thickness [m] |
| $\varphi_{mix}$ | Air–agent gas mixture gas concentration [kg/m$^3$] |
| $\omega_{max}$ | Maximum interface thickness [m] |
| $\varphi_{o_2}$ | O$_2$ mass density [kg/m$^3$] |
| $\sigma_{ij}$ | collisional cross section [$A$] |
| $\Omega_{ij}$ | Collision integral (dimensionless quantity) |

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
