# Peer review of "A Study on the Effect of O2 Diffusion on the Retention Time of Inert Agents"

_applsci, doi:10.3390/app10196694_

Round 1
Reviewer 1 Report
This is a very interesting and quite novel conceptual approach to improving the accuracy of retention time predictions. The considerations of the theoretical model are justified by research program results and the numerical study (CFD). However, due to that load of novelties it should be more precisely described.
- Firstly “of on” in the title. There are some doubts that two prepositions next to each other are too many. Also what is the meaning of the word “species” in this context; also in the text e.g. line 18 “O2 species”.
- There are many symbols. It would be easier for readers if you could gather all the symbols (and their explanations) at the beginning of the paper.
- FES gases (line 72) IG-01, IG-55, IG-541, and IG-100 – the material characteristics should be added e.g. IG-100 (Nitrogen).
- In the introduction there is a promise to define the interface slope, α – which is the fundamental physical value in the consideration. However, in Chapter 2 “Theoretical background” α is not mentioned. The formula (7) for α is given in the Chapter 4 “Results”. But before that, α is already used in the caption of Fig. 6. Apart from formula 7, the physical meaning of α should be described more in more detail due to the basic significance of the quantity.
- 6 caption: interface slope (a) and O2 concentration (b). In both pictures (a) and (b) there is only O2 concentration, the α is not present.
- 7. There is the question of dependent (y) and independent (x) variables: y(x). There is H(CO2). It is not that height depends on concentration, but opposite that concentration depends on height: CO2 (h). All seven pictures should be reconstructed.
- 8. The caption should be reconstructed exactly along with the information which the pictures will bring.
- 15. Similar to the remark 6 (Fig. 7). Time can not affect height, it is the opposite.
- Line 658: “the retention time is accelerated”. Time should not be accelerated, rather prolonged.
Author Response
Thank you for giving us the opportunity to submit a revised draft of our manuscript titled A study of the effect of O2 diffusion on the retention time in an inert agent. We appreciate the time and effort that you have dedicated to providing your valuable feedback on our manuscript. We have been able to incorporate changes to reflect most of the suggestions provided by you. We have highlighted the changes within the manuscript. Therefore, we ask that you review this manuscript again. Responses to all comments were written as attachments.Please see the attachment.

Reviewer 2 Report
This work is very well organized, it presents an adequate distribution of the work among the different sections. It presents a strong introduction, not forgetting in the same way the correct mathematical component with the description of the model to be studied, presenting a proper and correct section for numerical simulation. Likewise, it presents the relevant results of the work in a structured way and points out new directions to take into account. The subject is relevant and worthy of being published, section 4 with the name of the Results is very strong and clarifying the subject under study.
The only thing I have to ask for is in the description of the figures / tables that in the end to be uniform the final point must be presented.
Author Response
Thank you for giving us the opportunity to submit a revised draft of our manuscript titled A study of the effect of O2 diffusion on the retention time in an inert agent. We appreciate the time and effort that you have dedicated to providing your valuable feedback on our manuscript. We have been able to incorporate changes to reflect most of the suggestions provided by you. We have highlighted the changes within the manuscript. Therefore, we ask that you review this manuscript again.

Round 2
Reviewer 1 Report
From my point of view: OK!. No more remarks . Appreciation to authors for their great and effective effort to improve and clarify the questions. With great respect